# Characterization of a Unique Pair of Ferredoxin and Ferredoxin NADP^+^ Reductase Isoforms That Operates in Non-Photosynthetic Glandular Trichomes

**DOI:** 10.3390/plants13030409

**Published:** 2024-01-30

**Authors:** Joshua T. Polito, Iris Lange, Kaylie E. Barton, Narayanan Srividya, B. Markus Lange

**Affiliations:** Institute of Biological Chemistry and M. J. Murdock Metabolomics Laboratory, Washington State University, Pullman, WA 99164-7411, USA; josh.t.polito@gmail.com (J.T.P.); ilange@wsu.edu (I.L.); kaylie.barton@wsu.edu (K.E.B.); narayanan.srividya@wsu.edu (N.S.)

**Keywords:** ferredoxin, glandular trichome, methylerythritol phosphate, non-photosynthetic, terpene

## Abstract

Our recent investigations indicated that isoforms of ferredoxin (Fd) and ferredoxin NADP^+^ reductase (FNR) play essential roles for the reductive steps of the 2*C*-methyl-D-erythritol 4-phosphate (MEP) pathway of terpenoid biosynthesis in peppermint glandular trichomes (GTs). Based on an analysis of several transcriptome data sets, we demonstrated the presence of transcripts for a leaf-type FNR (*L-FNR*), a leaf-type Fd (*Fd I*), a root-type FNR (*R-FNR*), and two root-type Fds (*Fd II* and *Fd III*) in several members of the mint family (Lamiaceae). The present study reports on the biochemical characterization of all Fd and FNR isoforms of peppermint (*Mentha* × *piperita* L.). The redox potentials of Fd and FNR isoforms were determined using photoreduction methods. Based on a diaphorase assay, peppermint R-FNR had a substantially higher specificity constant (*k*_cat_/*K*_m_) for NADPH than L-FNR. Similar results were obtained with ferricyanide as an electron acceptor. When assayed for NADPH–cytochrome c reductase activity, the specificity constant with the Fd II and Fd III isoforms (when compared to Fd I) was slightly higher for L-FNR and substantially higher for R-FNR. Based on real-time quantitative PCR assays with samples representing various peppermint organs and cell types, the *Fd II* gene was expressed very highly in metabolically active GTs (but also present at lower levels in roots), whereas *Fd III* was expressed at low levels in both roots and GTs. Our data provide evidence that high transcript levels of *Fd II*, and not differences in the biochemical properties of the encoded enzyme when compared to those of Fd III, are likely to support the formation of copious amounts of monoterpene via the MEP pathway in peppermint GTs. This work has laid the foundation for follow-up studies to further investigate the roles of a unique R-FNR–Fd II pair in non-photosynthetic GTs of the Lamiaceae.

## 1. Introduction

Terpenoids constitute the largest class of plant specialized metabolites. Despite this chemical diversity, the biosynthesis of all terpene carbon skeletons proceeds as a modular process through condensation reactions between C_5_ prenyl diphosphate precursors, most commonly isopentenyl pyrophosphate (IPP) and dimethylallyl pyrophosphate (DMAPP) [1]. In plants, two spatially separated pathways are responsible for generating IPP and DMAPP: the mevalonate (MVA) pathway operates in the cytosol, the endoplasmic reticulum, and peroxisomes, while the enzymes of the 2*C*-methyl-D-erythritol 4-phosphate (MEP) pathway are localized to plastids [2]. The contribution of the MVA and MEP pathways to the formation of different terpene skeletons in plants can vary substantially by species, cell type, and as a response to environmental signals, and crosstalk between these pathways has been described in numerous experimental systems [3,4].

The formation of terpenoids in glandular trichomes (GTs) of peppermint (*Mentha* × *piperita* L.), which are specialized anatomical structures that synthesize and store leaf volatiles [5,6], involves the import of a carbohydrate precursor and conversion, primarily via central carbon metabolism and the MEP pathway, into monoterpenes (C_10_) and sesquiterpenes (C_15_) [7,8,9]. Microscopic investigations demonstrated that the filling of peppermint GTs is a rapid process, taking only 30 h, when compared to 18–25 d for leaf expansion [10,11]. This observation indicated that secretory-stage mint GTs must support an extraordinarily high carbon flux through the MEP pathway. An independent confirmation of the high-flux hypothesis was provided by a combination of metabolic engineering efforts (up- and down-regulation of MEP pathway genes) and various types of flux models for GTs in various species (positive correlation between gene expression levels and the accumulation of end products) [12,13,14,15,16,17,18].

The MEP pathway consists of seven enzymatic steps [19], starting with the conversion of D-glyceraldehyde 3-phosphate and pyruvate into 1-deoxy-D-xylulose 5-phosphate by 1-deoxy-D-xylulose 5-phosphate synthase (Figure 1). An isomerization and reduction catalyzed by 1-deoxy-D-xylulose-5-phosphate reductoisomerase generates MEP, the eponymous first committed intermediate of the pathway in bacteria. A condensation reaction facilitated by 4-diphosphocytidyl-2*C*-methyl-D-erythritol synthase and subsequent phosphorylation mediated by 4-(cytidine-5′-diphospho)-2*C*-methyl-D-erythritol kinase produce 4-diphosphocytidyl-2*C*-methyl-D-erythritol 2-phosphate [19]. Cleavage of the cytidine monophosphate moiety is followed by an intramolecular cyclization through catalysis by 2*C*-methyl-D-erythritol-2,4-cyclodiphosphate synthase. The last steps involve two successive elimination/reduction reactions catalyzed by 4-hydroxy-3-methylbut-2-enyl-diphosphate synthase (HDS) and 4-hydroxy-3-methylbut-2-enyl-diphosphate reductase (HDR), thus yielding IPP and DMAPP [19] (Figure 1).

HDS and HDR each possess a [4Fe-4S] iron–sulfur cluster to interact with an electron-donating system [20,21]. In photosynthetic cells, electrons from photon capture can be transferred, via ferredoxin (Fd), to HDS or HDR; in contrast, in non-photosynthetic cells, such as roots, an electron shuttle system from NADPH, generated through the oxidative pentose pathway [22], to ferredoxin-NADP^+^ reductase (FNR), to Fd, and finally to HDS or HDR has been proposed [23] (Figure 1). The primary role of Fd in photosynthesis is to donate electrons to FNR and ultimately for the photoreduction of NADP^+^ [24]. However, the donation of electrons by Fd has been demonstrated for many reductive reactions in plastids, including those catalyzed by HDS and HDR [25]. Conventional plant Fds are small proteins (~11 kDa) with a [2Fe-2S] cluster, coordinated by cysteine residues, which acts as the acceptor site for the transfer of one electron from photosystem I to an acceptor. Fd diffuses as an electron shuttle in the stroma between PS I (being reduced) and FNR (being oxidized). FNR is a monomeric protein (~35 kDa) localized to the stroma-facing side of the thylakoid membrane. It contains a single non-covalently bound flavin adenine dinucleotide (FAD) as prosthetic group. Two reduced Fd molecules sequentially transfer one electron each to FAD, and the two electrons are then transferred from FAD to NADP^+^. The Fd and FNR isoforms that are most highly expressed in photosynthetic cells (referred to as leaf-type) [26,27] have redox potentials that are consistent with the physiological direction of electron flow to regenerate NADPH (leaf-type Fds at –400 to –435 mV; leaf-type FNR at –342 to –356 mV, and the NADP^+^/NADPH couple at –320 mV) [28,29,30,31]. Crystal structural analyses revealed that favorable hydrophobic and charge–charge interactions at the interface of Fd and FNR bound in a binary complex allow for the placement of the [2Fe-2S] cluster (Fd) in close proximity to the FAD prosthetic group (FNR), thereby contributing to an efficient electron transfer [32,33].

*Fd* and *FNR* genes that are expressed at high levels in non-photosynthetic cells (encoding root-type isoforms of the enzymes) are distinct from those involved in photosynthesis [34,35]. At the biochemical level, root-type Fds are more efficiently reduced by NADPH, via FNR, than their leaf-type counterparts [36,37], which is reflected by redox potentials that are similar to that of the NADP^+^/NADPH couple (root-type Fds at –320 to –340 mV; root-type FNR at –337 mV) [30,31,38]. Kinetic assays with several FNR and Fd isoforms indicated that the strongest interactions occur between root-type FNR and root-type Fd, thereby facilitating electron flow in the NADPH–FNR–Fd cascade [37,39].

Our recent investigations indicated that the regeneration of reduced Fd plays a critical role in maintaining high flux toward terpenoid biosynthesis via the MEP pathway in peppermint GTs [17]. A re-analysis of transcriptome data sets from several members of the Lamiaceae provided evidence for the presence of transcripts for leaf-type FNR (*L-FNR*), leaf-type Fd (*Fd I*), root-type FNR (*R-FNR*), and two root-type Fds (*Fd II* and *Fd III*) [17,40,41]. The peptide sequence of one of the root-type Fds (Fd III) was closely related to typical root-type Fd isoforms and the corresponding gene had low expression levels in GTs; in contrast, the other root-type Fd (Fd II) of the Lamiaceae was part of a distinct, previously unrecognized, subclade of root-type Fds, where the corresponding genes were expressed at very high levels in GTs [17]. In this context, it is important to note that GTs of previously studied Lamiaceae contain only leucoplasts [42], relying mostly on non-photosynthetic metabolism, and one would thus expect root-type *Fds* to be expressed.

Based on these intriguing results, we hypothesized that Fd II of the Lamiaceae might have evolved as a GT-specific isoform of root-type Fds. For the present study, we thus embarked on the biochemical characterization of all FNR and Fd isoforms of peppermint. We also evaluated the transcript abundance related to these isoforms across multiple organs and tissue types, thus providing evidence that the R-FNR/Fd II pair likely plays a prominent role in peppermint GTs.

## 2. Materials and Methods

### 2.1. Plant Growth Conditions

Flats of peppermint plants (*Mentha × piperita*) were grown under greenhouse conditions: Sunshine Mix #1-Fafard-1P (Sun Gro Horticulture; Agawam, MA, USA); temperature 23–27 °C; humidity 60–75%; natural lighting supplemented with sodium vapor lamps to generate a consistent 16 h day/8 h night cycle; average light intensity during the day was 250 μE m^−2^ s^−1^; daily watering; fertilizer treatment once a week with Peters 20-20-20 (R.J. Peters Inc., Chesterfield, MI, USA).

### 2.2. Cloning of Fd and FNR Genes

Young leaves and GTs isolated from leaves using a previously described method [17] were frozen in liquid nitrogen, ground with a mortar and pestle, and total RNA was extracted using the RNeasy Plant Mini Kit (Cat. No. 74904, Qiagen, Germantown, MD, USA) following manufacturer instructions. DNase was applied on-column to remove genomic DNA. RNA quality was evaluated by agarose gel electrophoresis and spectrophotometry. First-strand cDNA was generated with Maxima Reverse Transcriptase (Thermo Fisher, Waltham, MA, USA) using oligo(dT) primers. First-strand cDNA was used as the template for PCR amplification of the coding sequences for *L-FNR*, *R-FNR*, *Fd I*, *Fd II*, and *Fd III* using Phusion Polymerase (Thermo Fisher, Waltham, MA, USA) and primers are listed in Appendix A. The purified PCR products were then ligated into the expression vector pSBET [43] using T4 Ligase (Thermo Fisher, Waltham, MA, USA), employing a sticky-end-cloning method [44]. The constructs were then transformed into *Escherichia coli* Top 10 cells (Invitrogen, Carlsbad, CA, USA). Colonies confirmed to contain the transgene by colony PCR and sequencing were chosen for plasmid preparations. Full-length open reading frames of *Fd II* and *Fd III* to be used for generating standard curves for real-time quantitative polymerase chain reaction (qPCR) were subcloned into a pGEM vector as previously described [41] (more details below).

### 2.3. Heterologous Expression of Genes in E. coli and Purification of Recombinant Proteins

*E. coli* BL21 (DE3) cells (Thermo Fisher, Waltham, MA, USA) were transformed with pSBET constructs by heat shock and cultured on Luria–Bertani (LB) agar plates containing 50 mg/mL kanamycin at 37 °C overnight. Individual colonies were used to inoculate 5 mL cultures of liquid LB medium containing the same concentration of kanamycin at 37 °C and shaken at 220 rpm for 8 h, which were then used to inoculate 150 mL cultures of the same medium that were grown under the same conditions for another 16 h. Cultures were then cooled on ice, induced by adding isopropyl β-D-1-thiogalactopyranoside (IPTG) (Gold Biotechnology, St. Louis, MO, USA) to a final concentration of 0.3 mM, and shaken at 16 °C for 24 h (200 rpm) before centrifuging at 3900× *g* at 4 °C for 10 min (Series 25, New Brunswick, ThermoFisher Scientific, Waltham, MA, USA). Cells were resuspended in their respective purification buffers (described below), placed on ice, and disrupted by sonication using an Ultrasonic Cell Disruptor (Virsonic 475, The VirTis, Inc., Gardiner, NY, USA) with a 3.2 mm microprobe operated at 20% output power for bursts of 3 × 15 s (45 s of cooling on ice between bursts). The resulting homogenate was centrifuged at 14,000× *g* for 30 min at 4 °C to remove solids.

For the functional evaluation of FNR isoforms, the cell pellet was resuspended in 10 mM Tris-Base buffer (RPI, Mt. Prospect, IL, USA), pH 8.0, containing 1 mM phenylmethylsulfonyl fluoride (PMSF) (Sigma-Aldrich Co., St. Louis, MO, USA), sonicated with three 15 s bursts and cooled on ice for 45 s between bursts, and centrifuged at 15,000× *g* for 30 min at 4 °C. The supernatant was fractionated with ammonium sulfate (FisherScientific, Fair Lawn, NJ, USA) between 40 and 70% saturation. The precipitate was pelleted at 14,000× *g* for 10 min at 4 °C and resuspended in 15 mL of the aforementioned cell resuspension buffer before applying the suspension to a gravity column (Bio-Rad, Hercules, CA, USA) containing 600 μL of Macro-Prep High Q Resin (Bio-Rad, Hercules, CA, USA) thoroughly equilibrated with the same buffer. FNR must be heavily diluted with buffer to successfully bind to the column resin at this step. The column was rinsed with 5 column volumes of the resuspension buffer to remove unbound protein. FNR was eluted with 100 mM NaCl before desalting and concentrating with a 30K Microsep Advance Centrifugal Device (Pall Corporation, Port Washington, NY, USA) at 4000× *g* for 30 min at 4 °C, with the buffer being exchanged with that required for each biochemical assay (described below).

The purification of Fd isoforms was based on previously described methods [36], but with considerable modifications. The *E. coli* cell pellet was resuspended in 50 mM Tris-Base buffer, pH 7.5, containing 100 mM NaCl and 1 mM PMSF, sonicated with three 15 s bursts and cooled on ice for 45 s between bursts, and centrifuged 15,000× *g* for 30 min at 4 °C. The supernatant was applied to a 1.5 cm diameter gravity column containing 1.5 cm (approximately 2.6 mL) diethylaminoethyl (DEAE) cellulose anion exchanger (medium mesh; Sigma-Aldrich Co., St. Louis, MO, USA) thoroughly equilibrated with the resuspension buffer. The column was washed with the same buffer until the eluate was clear. Fd was eluted with 50 mM Tris-Base buffer, pH 7.5, containing 700 mM NaCl. Ammonium sulfate was added up to 65% saturation. The precipitate was pelleted at 4000× *g* for 30 min at 4 °C. The supernatant containing Fd was applied to a gravity column containing 600 μL DEAE cellulose (medium mesh; Sigma-Aldrich Co., St. Louis, MO, USA) equilibrated with 50 mM Tris-Base buffer, pH 7.5, containing 65% ammonium sulfate and without NaCl. Fd was eluted with the same buffer containing 400 mM NaCl and without ammonium sulfate before desalting and concentrating with a Vivaspin-2 PES membrane 5K centrifugal concentrator (Vivaproducts, Littleton, MA, USA) at 4000× *g* for 60 min at 4 °C. The buffer was then exchanged with that required for each biochemical assay (described below).

Each purified protein was transferred into 1.5 mL tubes, covered with a layer of argon gas, kept on ice, and used for further analyses within 24 h. SDS-PAGE was used to examine the progress of the purification method’s development [45], with Precision Plus Protein Standard (Bio-Rad, Hercules, CA, USA) being used as a size reference ladder (Appendix A). Gels were stained with colloidal Coomassie Brilliant Blue [46] for 16 h and de-stained with water for 24 h at 23 °C. Overall protein concentration was determined by the Bradford protein assay [47].

### 2.4. Spectral Analyses

Absorbance spectra for concentration measurements were recorded with a Synergy H1 microplate reader (BioTek, Winooski, VT, USA). The concentration of FNR was determined by sodium dodecyl sulfate (SDS) denaturation and quantification of dissociated FAD in solution using the molar extinction coefficient (Ɛ_450_ = 11.3 mM^−1^ cm^−1^) [48] in a 96-well UV–Vis plate (Corning Inc., Corning, NY, USA). The concentration of Fd was established using the molar extinction coefficient for spinach Fd (Ɛ_422_ = 9.7 mM^−1^ cm^−1^) [49].

### 2.5. Enzyme Assays

For kinetic assays, a CLARIOstar Plus plate reader with a liquid injection module (BMG Labtech, Ortenberg, Germany) was used. The plate reader was maintained at 30 °C (thermal reading of the plate reader was most consistent at this temperature). All buffers were degassed for 30 min in a sonicating water bath to prevent the formation of bubbles during assays. Michaelis–Menten kinetic constants were calculated using a non-linear fit analysis in GraphPad Prism version 9.5.1 (Dotmatics, Boston, MA, USA) with least squares regression and without weighting or constraints. To determine the kinetic constants of FNR isoforms and monitor activity following purification (without testing for interaction with Fd), the ferricyanide diaphorase assay [50] was adapted for use with a 96-well plate. The buffer was 100 mM Tris-Base at pH 8.2. L-FNR and R-FNR were at concentrations of 0.05 µM. The starting concentration of NADPH was varied between 6 and 400 µM to determine the *K*_m_ of NADPH or at an unvaried 200 µM to determine the “acceptor” *K*_m_. The reaction was initiated by the injection of potassium ferricyanide (Sigma-Aldrich Co., St. Louis, MO, USA) into the well by the plate reader to a final volume of 200 μL and to a concentration of 1.0 mM for determining the *K*_m_ of NADPH or between 0.016 and 1.0 mM when varied for “acceptor” *K*_m_ measurements. The decrease in absorbance at 340 nm (NADPH) and at 420 nm (ferricyanide) was measured over time. The slope of the change in absorbance over time was converted into the micromolar rate of change using the extinction coefficients for NADPH (Ɛ_340_ = 6.22 mM^−1^ cm^−1^) and potassium ferricyanide (Ɛ_420_ = 1.02 mM^−1^ cm^−1^). No loss of activity was measured with either isoform of FNR within 24 h when stored at 4 °C, while assays were completed within <3 h.

To evaluate the interaction between Fd and FNR isoforms, the cytochrome c reduction assay [51], again modified and adapted for a 96-well plate format, was employed. The buffer was 50 mM Tris-Base at pH 7.5. The starting concentration of NADPH was 200 µM and the concentration of R-FNR was 0.025 µM and that of L-FNR was 0.05 µM. The starting concentration of each Fd isoform was varied between 0.3 and 20 µM to determine the *K*_m_ of each Fd isoform paired with either R-FNR or L-FNR. The concentration of oxidized cytochrome c (from equine heart, Sigma-Aldrich Co., St. Louis, MO, USA) dissolved in buffer was determined using the extinction coefficient at 410 nm (Ɛ_410_ = 106 mM^−1^ cm^−1^). The reaction was initiated by the injection of cytochrome c into the well by the plate reader to a concentration of 1.0 mM and a final volume of 200 μL. The increase in absorbance at 550 nm (reduction of cytochrome c) was measured over time. The slope of the change in absorbance over time was converted into the micromolar rate of change using the extinction coefficient for reduced cytochrome c (Ɛ_550_ = 28 mM^−1^ cm^−1^). Due to the temperature sensitivity of this assay, which causes degradation of initial kinetic velocity if reactants are left at room temperature for extended periods of time, all reaction mixes were kept on ice between measurements and equilibrated to the internal temperature of the plate reader before initiating the reaction (30 °C, as thermal reading of the plate reader was most consistent at this temperature). All isoforms of purified Fd (under argon gas) were unaffected by storage for up to 12 d at 4 °C or −20 °C. This far exceeded the amount of time in which the assays were completed, including those to determine kinetic constants (<1 d). The data reported here represent outcomes from 3–5 replicate assays.

### 2.6. Photoreduction to Obtain Redox Potentials

Previously published methods for the determination of electric potentials by photoreduction of flavins and iron–sulfur proteins with 5-deazariboflavin [52,53] were modified. 5-Deazariboflavin (Toronto Research Chemicals, Toronto, ON, Canada) was dissolved into 50 mM Tris-Base at pH 7.5, filtered to remove any undissolved solids, and the concentration determined based on the extinction coefficient (Ɛ_400_ = 12,500 M^−1^ cm^−1^) [54]. The final concentration was adjusted to 40 μM, the solution divided into 100 μL aliquots, and kept at −20 °C until later use. Loss of activity occurred when 5-deazariboflavin was left at 4 °C for longer than 24 h, but no significant loss of activity was detected when using frozen aliquots. The reaction buffer, 50 mM Tris-Base at pH 7.5, was made anaerobic by flash-freezing in liquid nitrogen and thawing under vacuum three times, with the atmosphere above the buffer being replaced with argon gas after each freeze–thaw cycle. Each protein isoform was placed in a 500 µL screw-cap quartz cuvette (Starna Cells Inc., Atascadero, CA, USA) with a rubber septum cap containing a total volume of 400 μL buffer, 1 μM 5-deazariboflavin, 10 μM ethylenediaminetetraacetic acid (EDTA) (Mallinckrodt Baker, Inc., Phillipsburg, NJ, USA), and either neutral red (NR; E_m_ = −325 mV), benzyl viologen (BV; E_m_ = −359 mV), or methyl viologen (MV; E_m_ = −446 mV) as redox reference dye (Sigma-Aldrich, St. Louis, MO, USA) [55,56]. NR and BV were used as the basis for calculating the redox potentials of R-FNR, L-FNR, Fd II, and Fd III, but Fd I had a lower potential and thus MV was used as an additional reference dye. When NR was used during the photoreduction of FNR isoforms, MV was included at a concentration of 0.01 μM to act as a single-electron mediator that did not contribute to the reduced spectrum. The concentrations of both the dye and the protein were adjusted so that the approximate maximum absorbance was 0.1 from 300–900 nm. Once all reactants had been added to the cuvette, argon gas was blown through a syringe into the cuvette septum to displace any remaining air, and syringe holes in the septum were sealed with silicone high-vacuum grease (Dow Corning, Midland, MI, USA). The cuvette was exposed to successive short periods (15 s to 1 min) of illumination with an Icon 2100 Lumen LED rechargeable floodlight (Harbor Freight, Calabasas, CA, USA), and absorbance spectra were recorded after each exposure using an Evolution 201 UV–Vis spectrophotometer (Thermo Fisher, Waltham, MA, USA). The EDTA–light reduction of 5-deazariboflavin results in the formation of a radical dimer with a redox potential of −800 mV, which is sufficient to reduce all FAD and iron–sulfur-cluster-containing proteins [52]. Isosbestic points of 390 nm for NR and 450 nm for BV and MV were determined by subjecting each dye separately to photoreduction (Appendix A), and the change in absorption over time represented the ratio of oxidized vs. reduced protein without influence of the dyes. The redox ratio for NR was measured at an absorbance of 550 nm and for BV and MV at 650 nm, which was determined to be outside the range of the absorbance spectrum of oxidized Fd and FNR. The baseline was corrected by subtraction of the average absorbance from 850–900 nm, where there was no substantial contribution from either the protein or dye being measured. The electric potential at equilibrium was determined using a simplified Nernst equation [57], accounting for the two-electron reduction of NR and the FNR-bound FAD cofactor and the single-electron reduction of BV, MV, and the Fd-bound iron–sulfur cluster. The data reported here represent outcomes of 3–5 replicate assays.

### 2.7. Tissue Collection for qPCR Assays

Five biological replicates were harvested for each of the following sample types: leaf segments with intact GTs, leaf segments with their GTs removed (“de-glanded” leaves), isolated GTs, and tap roots. All samples were collected before the emergence of flower buds. For gland cell isolation, leaves were separated into three developmental stages based on their leaf blade length (under 10 mm, 10–20 mm, 20–30 mm), representing different stages of leaf maturity (with the smallest leaves containing the highest density of actively secreting GTs) [10,11] (Appendix A). GTs were isolated from leaves using a previously described method [17], frozen in liquid nitrogen, and ground using an MM301 ball mill (Retsch, Haan, Germany) prior to RNA extraction. Root cuttings (rinsed and patted dry with a paper towel to remove soil) were frozen in liquid nitrogen and ground with a mortar and pestle prior to RNA extraction. The de-glanding of leaves was performed by adhesion of GTs to Scotch transparent tape (3M, St. Paul, MN, USA) prior to freezing with liquid nitrogen. The tips of large (>25 mm blade length) leaves were used for this process as the tips were extremely flat and it was thus easier to remove glands. The success of de-glanding leaves was confirmed using a dissecting microscope (WILD M3B, Leica, Wetzlar, Germany). Each leaf tip was de-glanded on one half of the leaf surface and separated into de-glanded and intact leaf samples. Tap roots and rhizomes (horizontally growing underground stems that send out roots) were collected separately. Peppermint is asexually propagated, and all plants are genetically identical; thus, biological replicates were grown from the same rootstock but in separate 60 × 30 cm flats. RNA was extracted and first-strand cDNA generated as described above.

### 2.8. qPCR Assays

Standard curves for absolute quantification were generated by performing qPCR with a range of concentrations of a BamHI-linearized pSBET plasmid (for *L-FNR*, *R-FNR*, and *Fd I*) or NcoI-linearized pGEM plasmid (for *Fd II* and *Fd III*) in accordance with previously established community guidelines [58]. Linearized plasmids were purified from agarose gels using the PureLink Quick Gel Extraction Kit (Invitrogen, Carlsbad, CA, USA). The DNA concentration was determined using a Synergy H1 plate reader and Take3 Micro-Volume plate (BioTek, Winooski, VT, USA). OligoCalc (version 3.27) [59] was used to calculate the copy number for each standard prior to dilution. A dilution series encompassing 30 to 3.0 × 10^7^ copies was prepared for each target gene.

An initial list of six candidate reference genes (Appendix A) was selected based on previous publications that analyzed reference gene candidates in *Salvia hispanica* L. (chia) and *Rosmarinus officinalis* L. (rosemary), both in the *Lamiaceae* family, and reported the stability of expression patterns under normal vegetative growth conditions and from both root and leaf tissue [60,61]. Primers used for qPCR measurements of gene expression for *Fd* and *FNR* isoforms (5 genes) and reference gene candidates (6 genes) are listed in Appendix A and were designed using the PrimerQuest tool (version 2.2.3) (Integrated DNA Technologies, Coralville, IA, USA) with coding sequences for each candidate gene [41]. Primers for *Fd II* and *Fd III* were designed to span the region representing the plastidial targeting sequence to capitalize on sequence divergence in this area.

Reactions were performed in triplicate (technical replicates) for each of the five biological replicates with the PowerTrack SYBR Green Master Mix (Cat. No. A46109, ThermoFisher Scientific Baltics UAB, Vilnius, Lithuania) as the qPCR reaction master mix, following the manufacturer instructions, with reactions being contained within 96-well Hard-Shell PCR Plates (Cat. No. HSP9601, Bio-Rad, Hercules, CA, USA). A CFX Connect Real-Time PCR Detection System (Bio-Rad, Hercules, CA, USA) was used as the qPCR thermal cycler, with data processed by CFX Manager 3.1 software (Bio-Rad, Hercules, CA, USA). The thermal cycler program was as follows: initial denaturing at 95 °C for 2 min, then 40 cycles of 95 °C for 15 s and 60 °C for 30 s. All experimental gene primers were confirmed to demonstrate no off-target amplification between similar isoforms and had an efficiency of 90% or above. First-strand cDNA was diluted 1:10 for all tissue types. Reference genes were evaluated for suitability using NormiRazor (original version downloadable at https://norm.btm.umed.pl (accessed on 28 January 2024)) [62], with three genes ultimately being selected: α-tubulin (*α-Tub*), protein phosphatase 2A (*PP2A*), and eukaryotic translation initiation factor 3 subunit E (*EIF3E*). Expression levels of all experimental genes were normalized by multiplying the C_q_ by the geometric mean [63] of the stability factors determined by NormiRazor for each of the three reference genes in each tissue type. The normalized C_q_ value was employed to calculate the copy number based on each gene’s standard curve. The MIQE guidelines [64] served as the basis for reporting our qPCR data.

## 3. Results and Discussion

### 3.1. L-FNR and R-FNR of Peppermint Are Archetypal Leaf and Root Isoforms, Respectively

cDNAs for the *L-FNR* and *R-FNR* genes, the identification of which we described previously [17], were subcloned into the pSBET expression vector [43] and functionally expressed in *E. coli* BL21 (DE3) cells. The corresponding recombinant proteins were purified separately by the successive use of ammonium sulfate fractionation, anion exchange chromatography, and size exclusion spin columns, thus resulting in highly enriched fractions that gave rise to 35 kDa bands on an SDS-PAGE gel. Purified FNR isoforms had absorbance spectra typical of FAD-containing proteins, with λ_max_ at 392/460 nm for R-FNR and 380/456 nm for L-FNR; following denaturation with SDS, λ_max_ shifted to 375/450 nm, which is characteristic for a dissociated FAD cofactor (Appendix A). An NADPH-dependent diaphorase assay was employed to assess the capacity of these purified FNRs to facilitate electron transfer reactions [50]. In this assay, the transfer of electrons from NADPH to an acceptor (potassium ferricyanide in this study), mediated by FNR, is employed, which is recorded as a decrease in absorbance at 340 nm (NADPH oxidation) and 420 nm (ferricyanide reduction). Michaelis–Menten curves were the basis for calculating kinetic constants by varying the concentrations of NADPH (Appendix A) or potassium ferricyanide (Appendix A) in the assay. *K*_m_^NADPH^ was determined as 178 µM for L-FNR and 59 µM R-FNR, while *k*_cat_^NADPH^ was obtained as 48 s^−1^ for L-FNR and 266 s^−1^ for R-FNR (Table 1). The resulting specificity constant (*k*_cat_/*K*_m_) for NADPH was 17-fold higher for R-FNR than for L-FNR. The trend for the binding affinity of ferricyanide was the opposite that for NADPH, with a lower *K*_m_ of 76 µM for L-FNR and a *K*_m_ of 190 µM for R-FNR, while the rate constant (*k*_cat_) was higher for R-FNR (191 s^−1^) than for L-FNR (52 s^−1^). The specificity constant (*k*_cat_/*K*_m_) for ferricyanide was 1.5-fold higher for R-FNR than for L-FNR (Table 1). While the absolute values for kinetic constants can vary based on the assay conditions (e.g., salt concentration, temperature, pH, or electron acceptor) [65], the trends when comparing properties of FNR isoforms tend to be consistent [50]. The specificity constant for NADPH (*k*_cat_/*K*_m_), as reported by others for maize (*Zea mays* L.) isoforms, was considerably higher for R-FNR than for L-FNR [37,66], consistent with our data. Based on comparisons of sequence characteristics and crystal structures, it has been suggested that the higher binding affinity of R-FNR might be due to interactions of the 2′-phospho-AMP portion of NADP(H) with positively charged side chains of active site residues (K238 and R240) that correspond to neutral or negative charges in L-FNR (T238 and E/A240); in addition, the pyrophosphate moiety of NADP(H) might interact more strongly with R116 (R-FNR) than K116 (L-FNR) (numbering of mature spinach L-FNR used as reference) [30]. The same differences are present in the R-FNR and L-FNR sequences of peppermint (Appendix A), indicating that the peppermint genome encodes archetypical members of the root- and leaf-type FNRs.

### 3.2. Peppermint R-FNR Interacts Favorably with Root-Type Fd Isoforms, While L-FNR Has Lower Specificity Constants with All Fd Isoforms

cDNAs corresponding to the genes coding for peppermint Fd I, Fd II, and Fd III, which had been identified previously [17], were subcloned into the pSBET expression vector [43] and functionally expressed in *E. coli* BL21 (DE3) cells. The recombinant proteins were purified separately by sequential anion exchange chromatography, ammonium sulfate fractionation, a second step of anion exchange chromatography, and concentration over size exclusion spin columns, thus resulting in highly enriched fractions that gave rise to bands of roughly 15 kDa on an SDS-PAGE gel. All peppermint Fd isoforms had absorbance spectra resembling typical iron–sulfur cluster proteins with λ_max_ of 330/420 nm (Appendix A). Static interactions between FNR and Fd were studied using a modified version of the NADPH-dependent cytochrome c reductase assay [50]. During the reaction, electrons are transferred from NADPH to FNR, on to Fd, and ultimately from Fd to cytochrome c, which is shown by an increase in absorbance at 550 nm (cytochrome c reduction) and a decrease at 340 nm (NADPH oxidation), and then used to calculate kinetic constants (Appendix A). Peppermint L-FNR had a slightly lower *K*_m_ value for leaf-type Fd I (2.3 µM), when compared to root-type Fd II or Fd III (*K*_m_ of 2.8 and 2.9 µM for both isoforms) (Table 2). In the same assays with R-FNR, the *k*_cat_ values for Fd II and Fd III were significantly higher than that for Fd I (2.5- and 2.3-fold, respectively). The same trend was observed in the differences of the specificity constants (*k*_cat_/*K*_m_) (Table 2). Peppermint R-FNR had a > 2-fold higher *K*_m_ value for Fd II and Fd III (1.9 and 2.1 µM, respectively) than for Fd I (*K*_m_ of 3.6 µM). The rate constant was also higher for Fd II and Fd III (*k*_cat_ of 428 s^−1^ for both Fd II and Fd III) compared to Fd I (*k*_cat_ of 281 s^−1^). This was reflected in considerably higher catalytic specificity constants for Fd II and Fd III (221 and 189 µM^−1^ s^−1^, respectively) compared to that of Fd I (*k*_cat_/*K*_m_ of 78 µM^−1^ s^−1^) (Table 2). The kinetic efficiency was significantly (2-fold) higher for R-FNR–Fd II than for the L-FNR–Fd II pair; it was also higher for R-FNR–Fd III compared to the L-FNR–Fd III pair (also 2-fold). Earlier work reported a 3.3-fold difference for the R-FNR–Fd III versus L-FNR–Fd III comparison of the specificity constant (*k*_cat_/*K*_m_) [37], which is very similar to the difference observed in the present work (Table 2). In summary, our data are consistent with those of earlier reports, demonstrating that R-FNR has a substantially higher catalytic efficiency with root-type Fd (when compared with leaf-type Fd), whereas L-FNR does not differentiate as clearly between leaf-type and root-type Fd isoforms (demonstrated for both maize and *Arabidopsis thaliana* L. enzymes) [31,37,39]. The combination of comparatively low *K*_m_ and high *k*_cat_ values for the complex of R-FNR with root-type Fd isoforms bears the potential for favorable electron transfer in reductive reactions in non-photosynthetic cell types (where these isoforms are prevalent), but the physiological relevance remains to be determined.

The evaluation of crystal structures and subsequent site-directed mutagenesis efforts previously identified residues for the interaction and electron transfer between L-FNR and leaf-type Fd isoforms. These data suggested that basic residues with positively charged side chains (in particular K85, K88, and K91; numbering of the mature spinach L-FNR used as reference) aid in the formation of a 1:1 complex through electrostatic interactions with negative charges on the Fd surface [67,68,69,70]. When the maize R-FNR residues equivalent to K85, K88, and K91 of L-FNR (N, A, and N, respectively) were mutated to those present in L-FNR, the *K*_m_ value for the interaction with leaf-type Fd decreased 6.8-fold, indicating that these residues indeed play an important role in fostering tight interactions between the proteins in the complex [39]. Similar results were reported for D300 of maize R-FNR (corresponds to F in L-FNR). It was also shown that the exchange of the residues S154 and I157 of maize R-FNR (mutated to V and E, respectively, which are the residues present in L-FNR) increased the *K*_m_ value for the interaction with root-type Fd 6.2-fold, which would appear to suggest that these residues are important for differentiating between leaf-type and root-type Fd isoforms. The residues mentioned above are conserved among root-type FNRs, including the R-FNR of peppermint characterized in the present study (Appendix A).

Sequence differences in Fd isoforms were also explored to assess which residues are involved in the tighter binding of root-type Fd to R-FNR (compared to the binding of leaf-type FNR). Based on an analysis of a series of leaf-type Fd mutants, in which certain residues were exchanged with those present in root-type Fd, a septuple mutant of maize Fd I (D34S, S43A, S46T, Q61G, T96Y, ΔG97, ΔA98 (the last two modifications are deletions)) was generated that had the same *K*_m_ value as that of maize Fd III, thus providing evidence for the role of these residues in the differential binding characteristics of the wild-type Fd isoforms to R-FNR [33,39]. Fd I of peppermint had the sequence characteristics of a typical leaf-type Fd, while the attributes of Fd II and Fd III of peppermint agree with those of a root-type Fd (Appendix A), which is consistent with our biochemical data that demonstrated a tighter association of R-FNR with Fd II and Fd III (compared to Fd I).

### 3.3. The Redox Potentials of Peppermint Fd I and Fd II/Fd III Are Typical for Leaf-Type and Root-Type Fd Isoforms, Respectively

A stepwise anaerobic photoreduction of the FAD prosthetic group of FNR or the [2Fe-2S] cluster of Fd was used to determine redox potentials of peppermint isoforms of these enzymes (Figure 2). Peppermint Fd I had a redox potential of –422 mV, which is similar to leaf-type Fds of Arabidopsis (Fd I at –425 mV and Fd II –433 mV) and maize (Fd I at –401 mV and Fd II –423 mV) [29,30,31]. Peppermint Fd I had a significantly lower redox potential than the FAD moiety of leaf-type FNRs (peppermint L-FNR at –338 mV and maize L-FNR at –356 mV) [30], consistent with the physiological direction of electron flow in photosynthetic cells from Fd to FNR and ultimately to NADP^+^ (Figure 3). The redox potentials of peppermint Fd II (–344 mV) and Fd III (–346 mV) were close to those of peppermint R-FNR (–324 mV) and NADPH (–320 mV) which would be suitable for electron transport from NADPH to Fd via FNR, as expected for non-photosynthetic cells. The redox potentials of maize Fd III (–321 mV) and maize R-FNR (–337 mV) [30] were similar to the corresponding peppermint isoforms (Figure 3).

Taken together, the biochemical data support the classification of peppermint Fds as leaf-type (Fd I) and root-type (Fd II and Fd III) isoforms. However, we did not observe substantial differences in the properties of Fd II and Fd III, which begs the question why the peppermint genome encodes two root-type Fd isoforms, whereas only one root-type isoform is found in Arabidopsis and maize. We therefore hypothesized that the occurrence of two root-type isoforms in peppermint (and other members of the Lamiaceae) might facilitate high expression levels of Fd II in glandular trichomes, while Fd III might be dominant in roots (both contain non-photosynthetic cell types).

### 3.4. Fd II Is Expressed in Non-Photosynthetic Cells of Peppermint, with Particularly High Transcript Abundance in Secretory-Stage Glandular Trichomes

A qPCR study was designed to evaluate expression patterns of the genes that code for Fd and FNR isoforms in peppermint. Taproots were separated from rhizomes and harvested (Figure 4). Different leaf samples were harvested based on size (blade length < 10 mm, 10–20 mm, and 20–30 mm), representing different developmental stages (Figure 5A–C). Glandular trichomes were isolated from leaves in a bead beater chamber filled with a viscous buffer and glass beads (Figure 5E,F). Transparent Scotch tape was employed to remove glandular trichomes from leaves, while keeping the remainder of the leaf intact (“de-glanding”), which allowed us to investigate leaves with very few glandular trichomes as a reference (bead beating shatters samples into very small fragments that cannot be used as a source of de-glanded leaves) (Figure 6). *L-FNR* was expressed primarily in leaves (43,024 transcripts per ng RNA), with only background expression levels in roots and gland cells isolated from leaves with <20 mm blade length (≤100 transcripts per ng RNA) (Figure 7). A slightly higher *L-FNR* transcript abundance (360 transcripts per ng RNA) was detected in gland cells isolated from older leaves (20–30 mm blade length) but these preparations had a slightly greenish color (Figure 5D), indicating that some highly fragmented leaf material contaminated the gland cell preparation. The expression levels in de-glanded leaves were higher than intact leaves (48,160 transcripts per ng RNA), consistent with a localization to leaves but not glands. The trends were the same for *Fd I*: high expression in leaves (185,295 transcripts per ng RNA), even higher expression in de-glanded leaves (196,093 transcripts per ng RNA), low expression levels in taproots and gland cells isolated from leaves with <20 mm blade length (≤2400 transcripts per ng RNA), and slightly higher transcript abundance in gland cells isolated from older leaves (20–30 mm blade length) (12,408 transcripts per ng RNA) (Figure 7). The presence of *FdI* transcript in GTs isolated from older leaves is likely due to contamination from broken photosynthetic leaf cells, as indicated by the greenish color of RNA preparations (Figure 5). The gene expression patterns of peppermint *Fd I* and *L-FNR* are consistent with those reported for the genes coding for the leaf-type *Fd* and *FNR* isoforms in maize and Arabidopsis, which have been demonstrated to be expressed primarily in photosynthetic cell types [35,65,71,72,73].

In contrast, peppermint *R-FNR* was expressed at very low levels in leaves and de-glanded leaves (≤1100 transcripts per ng RNA), low levels in roots (4932 transcripts per ng RNA), and at generally high levels in gland cells, with a gradual leaf age-dependent decrease (88,088 to 73,678 transcripts per ng RNA) that corresponds to the proportion of actively secreting GTs [10,11]. A similar expression pattern was observed for *Fd II* but with dramatically higher transcript abundance in gland cells (309,855 transcripts per ng RNA in preparations obtained from leaves of <10 mm in blade length) (Figure 7). *Fd II* expression was also elevated in roots (20,152 transcripts per ng RNA) when compared to the low levels in leaves and de-glanded leaves (1677 and 2105 transcripts per ng RNA, respectively). Transcript abundance of *Fd III* was comparable in roots and all gland cell samples (10,000 to 20,000 transcripts per ng RNA) but much lower than *Fd II* (Figure 7). The expression of *Fd III* in leaves and de-glanded leaves was even lower (2118 and 2685 transcripts per ng RNA). While previously published transcriptome data indicated high expression levels for *Fd II* and low transcript abundance of *Fd III* in GTs of members of the Lamiaceae [17], this is the first qPCR study to investigate the gene expression levels of all Fd and FNR isoforms quantitatively across multiple cell types, including GTs. Our data indicate that *Fd II* is the dominant isoform expressed in peppermint GTs, which might ensure that a sufficient quantity of the corresponding enzyme can be made to support the high flux through the MEP pathway in these non-photosynthetic cells. In this context, it should be noted that there is generally a good correlation between transcript and protein abundance in actively secreting glandular trichomes, as indicated by a meta-analysis of dozens of transcriptome and proteome studies [74]. A careful Fd and FNR isoform analysis at the protein level across peppermint tissue types, to experimentally test these assumptions, would still be a worthwhile future endeavor.

## 4. Conclusions

The current study investigated the occurrence of two root-type Fd isoforms in members of the Lamiaceae. Using peppermint as a model, we demonstrated that the biochemical properties of Fd II and Fd III, in terms of binding NADPH and associating tightly with R-FNR, were very similar. The redox potentials of these isoforms were almost identical and well suited for a role in the transfer of electrons from NADPH to FNR, from there to Fd, and then on to a redox-active enzyme in non-photosynthetic cell types. However, the expression patterns of the corresponding genes were very different, with *Fd II* being expressed very highly in GTs (but also present at lower levels in roots), whereas *Fd III* was expressed at low levels in both roots and GTs. Our data indicate that both Fd II and Fd III likely contribute to reductive reactions in roots but that Fd II is the dominant isoform in GTs. Furthermore, we provide evidence that high transcript abundance of Fd II, and not differences in the biochemical proper-ties of the encoded protein when compared to Fd III, is supporting the high flux through the HDS and HDR enzymes of the MEP pathway in GTs. This hypothesis can now be tested in follow-up studies, outside the scope of the current investigation, to determine if knocking down *Fd* genes individually and cell-type-specifically has an effect on products derived from the MEP pathway.

## Figures and Tables

**Figure 1 plants-13-00409-f001:**
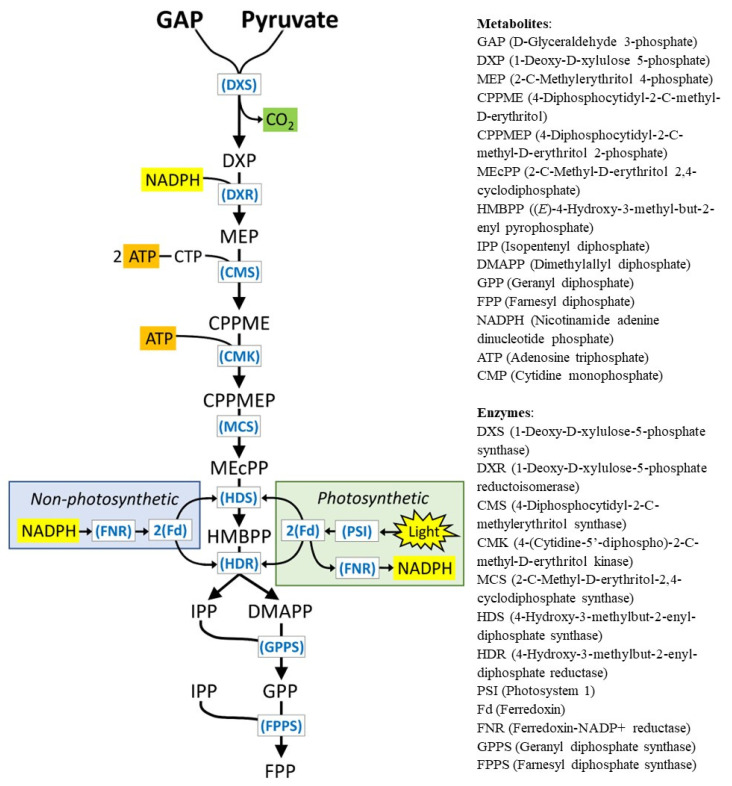
The MEP pathway in plants. The reduction of HDS and HDR by Fd to produce terpene precursors (IPP and DMAPP) can use NADPH as a source of electrons in non-photosynthetic GTs or light as a source of electrons in photosynthetic GTs. ATP as cellular energy currency, NADPH as reduced electron carrier, and CO_2_ as byproduct of the MEP pathway are highlighted (orange, yellow, and green, respectively).

**Figure 2 plants-13-00409-f002:**
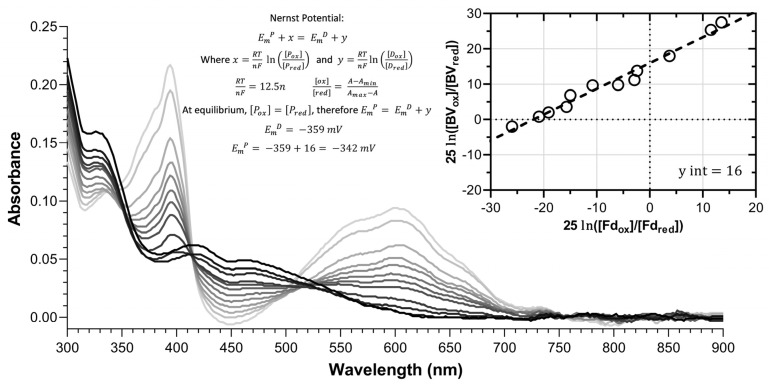
Exemplary photoreduction data showing the light-induced change in the absorbance spectrum over time for a cuvette containing Fd II and benzyl viologen (BV). Time-resolved spectra are shown with a gradient from black to light gray, with black being the most oxidized and gray being most reduced. The inset equations under “Nernst Potential” (top middle) were used to calculate the redox potential. The ratio of oxidized to reduced protein and dye is measured by monitoring the decrease in absorbance at 450 nm for Fd and the increase at 650 nm for BV. The inset graph (top right) shows the Nernst plot used to obtain the y-intercept (data points indicated as hollow circles; trendline shown as broken line), which is the difference in potential between the dye and the protein. Equation variables: Em = redox potential; R = ideal gas constant; T = temperature in Kelvin; n = number of electrons transferred; F = Faraday’s constant; A = absorbance; abbreviations: P = protein; D = dye; ox = oxidized; red = reduced; BV = benzyl viologen; Fd = ferredoxin.

**Figure 3 plants-13-00409-f003:**
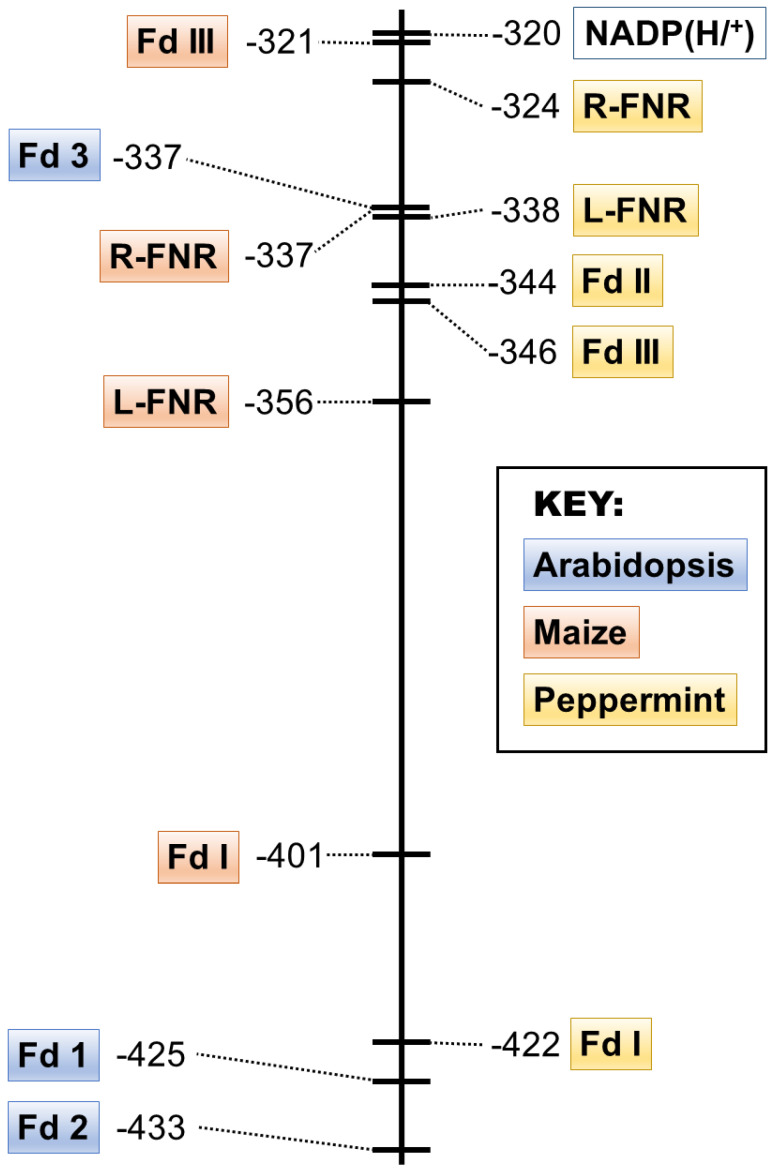
Redox potential (mV) of Fd and FNR isoforms found in *Arabidopsis thaliana* (highlighted in blue) [31], *Zea mays* (maize; highlighted in orange) [30], and *Mentha × piperita* (peppermint; highlighted in yellow) (this work). The redox potential of the NADP(H/^+^) couple is shown for comparison.

**Figure 4 plants-13-00409-f004:**
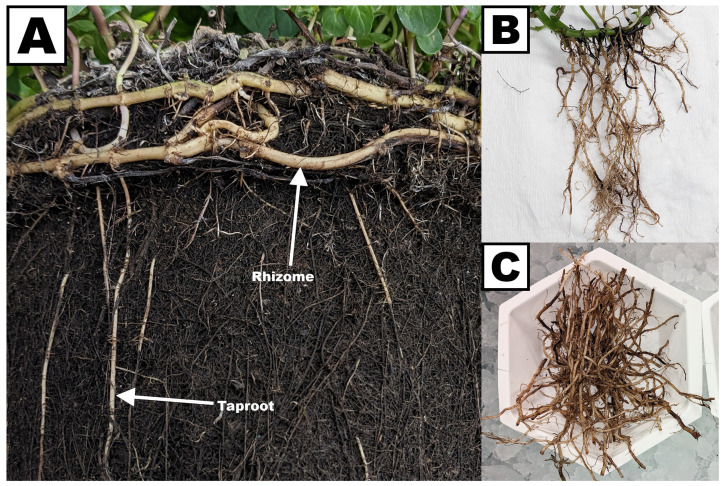
Preparation of root samples for qPCR. (**A**) Photo of subsurface rhizomes and roots of a typical peppermint flat. Rhizomes tend to spread along the soil surface and contain a mixture of green photosynthetic and white non-photosynthetic cells. Taproots emerge from rhizomes and penetrate the soil vertically and consist only of white non-photosynthetic cells. (**B**) Washed rhizomes and taproots. (**C**) Taproots separated from rhizomes (all non-photosynthetic) prior to grinding in liquid nitrogen for RNA extraction.

**Figure 5 plants-13-00409-f005:**
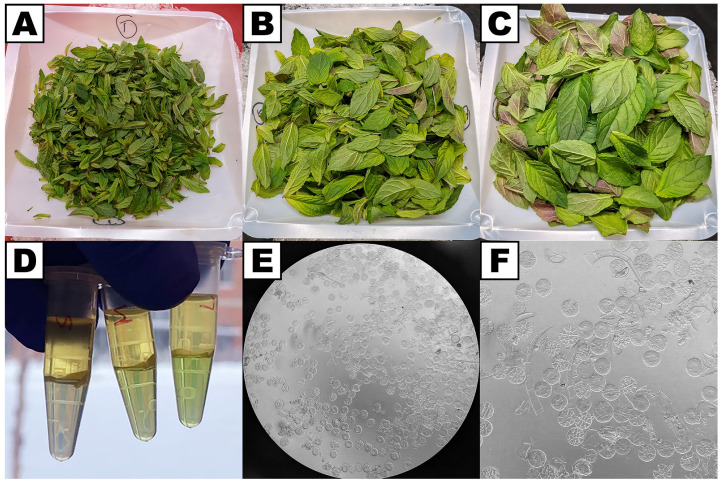
Leaves collected from peppermint prior to the gland prep protocol. (**A**) Under 10 mm leaf blade length, (**B**) 10–20 mm leaf blade length, (**C**) 20–30 mm leaf blade length. (**D**) Aqueous/chloroform extraction of RNA from glands of leaves of different ages (same order as in panels (**A**–**C**)), showing increased chlorophyll content (green color) resulting from increased leaf fragment contamination in glands from older leaves. (**E**) Microscopy image of gland cells at 40× magnification. (**F**) The same image as in panel (**E**) but zoomed in.

**Figure 6 plants-13-00409-f006:**
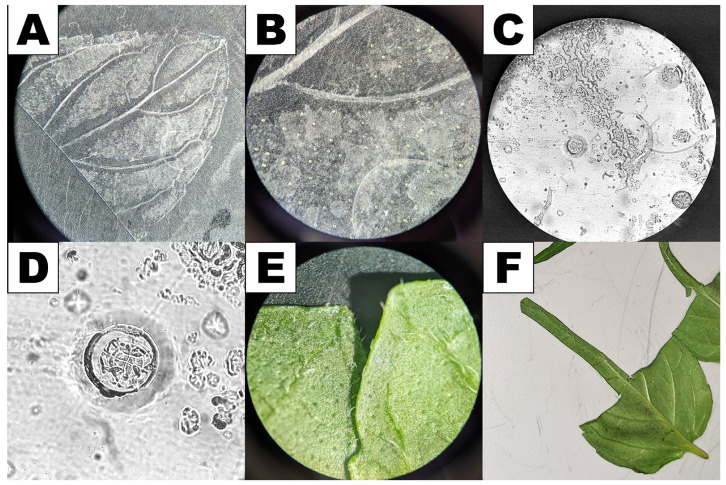
Preparation of de-glanded leaf samples for qPCR. (**A**) Imprint of peppermint leaf left on transparent Scotch tape, showing outline of leaf veins and lack of imprint near veins. (**B**) Closer view of leaf imprint on tape showing individual glands removed from leaf, especially where leaf is flat (away from veins). (**C**) Closer view of leaf imprint on tape showing multiple types of trichomes having been removed. (**D**) Close-up of a secretory cell disk (from panel (**C**)), demonstrating removal of intact gland. (**E**) Comparison of a de-glanded leaf (left) to an intact leaf (right). Glands appear as shiny bubbles on the leaf surface. When removed, a dark pit is left in its place. (**F**) Leaf segments after half has been harvested intact and half has been removed following de-glanding.

**Figure 7 plants-13-00409-f007:**
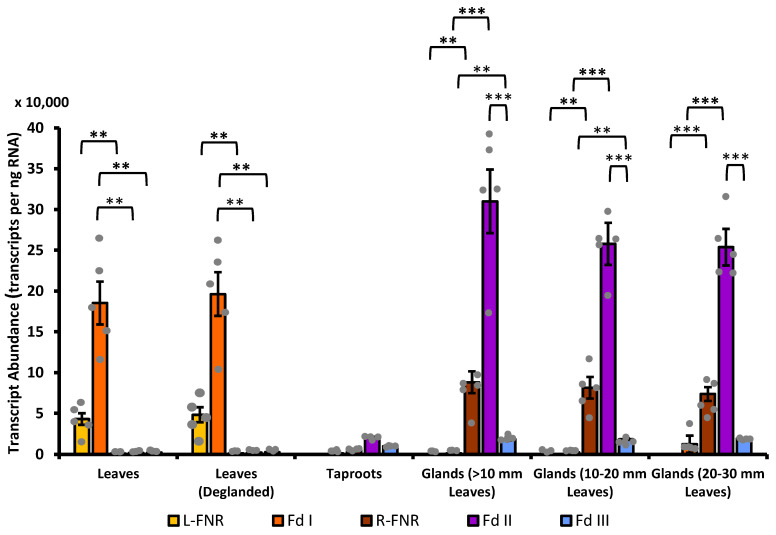
qPCR data to quantify the expression levels of L-FNR and Fd I (photosynthetic isoforms; orange and red bars, respectively) and R-FNR, Fd II, and Fd III (non-photosynthetic isoforms; brown, purple, and blue bars, respectively) measured in leaves, de-glanded leaves, taproots, and glands harvested from leaves at different stages of maturity. Columns indicate average values of five biological replicates. Bars show standard error, with individual data points being represented as gray dots. The statistical significance (*p*-value) of pairwise comparisons, based on a two-tiered Student *t*-test, is indicated by asterisks: **, ≤0.01; ***, ≤0.001.

**Table 1 plants-13-00409-t001:** Michaelis–Menten kinetic constants for the FNR-dependent ferricyanide reduction.

	*k*_cat_^NADPH^ (s^−1^)	*K*_m_^NADPH^ (μM)	*k*_cat_/*K*_m_^NADPH^ (μM^−1^s^−1^)	*k*_cat_^acceptor^ (s^−1^)	*K*_m_^acceptor^ (μM)	*k*_cat_/*K*_m_^acceptor^ (μM^−1^s^−1^)
R-FNR	266.6 ± 12.0	58.6 ± 8.4	4.5 ± 0.6	191.1 ± 15.4	189.8 ± 42.8	1.0 ± 0.2
L-FNR *	48.0 ± 10.5	178.0 ± 23	0.3 ± 0.1	51.7 ± 1.6	76.4 ± 8.5	0.7 ± 0.1

* = Photosynthetic isoform. Averages and standard errors for 3–5 replicates are given.

**Table 2 plants-13-00409-t002:** Michaelis–Menten kinetic constants for the FNR–Fd-dependent cytochrome c reduction.

	*k*_cat_ (s^−1^)	*K*_m_^Fd^ (µM)	*k*_cat_/*K*_m_^Fd^ (µM^−1^s^−1^)	*k*_cat_ (s^−1^)	*K*_m_^Fd^ (µM)	*k*_cat_/*K*_m_^Fd^ (µM^−1^s^−1^)
	R-FNR			L-FNR *		
Fd I *	281.2 ± 19.6	3.6 ± 0.6	78.3 ± 10.4	125.2 ± 2.4	2.3 ± 0.1	54.2 ± 2.6
Fd II	428.2 ± 9.2	1.9 ± 0.1	221.8 ± 12.7	304 ± 7.8	2.8 ± 0.2	108.9 ± 7.8
Fd III	428 ± 10.8	2.3 ± 0.2	189.4 ± 19.5	285.6 ± 6.6	2.9 ± 0.2	98.1 ± 7.2

* = Photosynthetic isoform. Averages and standard errors for 3–5 replicates are given.

## Data Availability

The original contributions presented in the study are included in the article/Appendix A. Further inquiries can be directed to the corresponding author.

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
