# Peer review of "Characterization of a Unique Pair of Ferredoxin and Ferredoxin NADP+ Reductase Isoforms That Operates in Non-Photosynthetic Glandular Trichomes"

_plants, 2024, doi:10.3390/plants13030409_

Round 1
Reviewer 1 Report
Comments and Suggestions for Authors
The report by Polito and colleagues investigates the role played by various leaf and root ferredoxins plus ferredoxin reductases within members of the mint family of plants in the MEP pathway involved in the fueling the formation of typical monoterpenes of glandular trichomes high transcript levels of Fd II are responsible for the high level of formation of monoterpenes in glandular trichomes rather than the biochemical properties of the enzyme product. This conclusion is supported by the almost identical biochemical properties of Fd II and Fd III.
The preferential expression of one transcript isoform in a cell type or organ is typical of the mechanisms involved in the regulation of one biochemical role over another and has been described in the past for different biochemical processes. In this context it would have been very nice if experiments suppressing Fd II expression in glandular trichomes could have been performed in order test this interesting hypothesis, since one would expect that diminishing MEP pathway flux should lead to a decline of monoterpene accumulation. The availability of published mint transformation methods could have permitted such tests, although this would be a time-consuming experiment to perform.
The authors do suggest in their conclusions that knocking down of FD genes was outside the scope of the present investigation and might be subject for future studies to further test the hypothesis.
The report is well written, and the data presentation is clear and conclusive to test if a GT-specific isoform of Fd II of root-type Fds that led to the conclusion that high transcript expression of Fd II was responsible for the biochemical phenotype observed.
Questions:
1. Explain why an extensive purification protocol was required for recombinant Fd isoforms? Does tagging for affinity purification not work?
2. Were the purified isoforms sufficiently pure for determination of biochemical properties?
3. While expression of Fd II is many fold higher than that of Fd I, there is still significant expression of Fd I in glandular trichomes. Please provide some information of what Fd 1 might be doing in this organ and in relation presumably to the presence of chlorophyll-lacking plastids.
4. Another valuable approach would have been to monitor the actual levels of individual Fd isoform proteins to see if their abundance followed their transcript abundance. It is not clear to this reviewer that transcript abundance always translates into protein abundance, even of this heretical from the viewpoint of molecular biologists.
Author Response
Responses to Reviewer Questions and Comments:
1. Explain why an extensive purification protocol was required for recombinant Fd isoforms? Does tagging for affinity purification not work?
The preparations used on the assays we describe in our manuscript need to contain the target enzyme at very high purity. Nickel affinity purifications alone did not yield Fd of sufficient purity and additional purification steps were thus implemented.
2. Were the purified isoforms sufficiently pure for determination of biochemical properties?
There is no assay that unequivocally demonstrates that the purity of Fd and FNR enzymes is sufficient so that unintentional impacts from the presence of impurities can be excluded. To show the outcomes of purification protocols, SDS-PAGE results have been added as Supplementary Materials.
3. While expression of Fd II is many fold higher than that of Fd I, there is still significant expression of Fd I in glandular trichomes. Please provide some information of what Fd 1 might be doing in this organ and in relation presumably to the presence of chlorophyll-lacking plastids
Thank you for the suggestion. The revised version of the manuscript contains an expanded discussion to address the presence of FdI transcript in glandular trichomes.
4. Another valuable approach would have been to monitor the actual levels of individual Fd isoform proteins to see if their abundance followed their transcript abundance. It is not clear to this reviewer that transcript abundance always translates into protein abundance, even of this heretical from the viewpoint of molecular biologists.
We agree with the reviewer that transcript abundance does not always translate into protein abundance (although there is generally a very high level of correlation in specialized cell types). With that in mind, quantifying the protein quantities of different Fd and FNR isoforms in different cell types and organs would have been highly informative. However, obtaining such data is not trivial as there are no antibodies that have been demonstrated to work with the peppermint enzymes and are able to differentiate the different isoforms. An alternative to the immunological quantitation would have been quantitative proteomics, for example with aqua peptides, but these experiments are essentially a study of their own (considering the enormous amount of work needed for methods development and obtaining the plant samples). More discussion of this suggestion has been added to the revised manuscript.
Reviewer 2 Report
Comments and Suggestions for Authors
This is a well designed, biochemical study on the electron transfer proteins involved in mint essential oil production in glandular trichomes through the MEP pathway. The authors present a logical, unified conclusion (i.e., tissue specific transcript expression and not biochemical differences supports high flux through the peppermint oil pathway) that is well supported by the data presented. The spectroscopy methods are rigorous and placed in context of similar research in other plants systems.
I have no major concerns. Minor observations are as follows:
1. Authors state “…crosstalk between these pathways has been described in numerous experimental systems (Hemmerlin et al., 2012).” Can additional references (preferably from the last 5 years) be provided to support this statement?
2. Line 55, misspelling in Amelunxen, 1965; Amelunxes et al., 1969
3. At line 70, authors state “An isomerization and reduction catalyzed by 1-deoxy-D-xylulose-5-phosphate reductoisomerase generates MEP, the eponymous first committed intermediate of the pathway.” Could the authors qualify this statement as perhaps “…first committed intermediate…in bacteria.”? There is good support DXP is in fact the first committed intermediate in plants as vitamin B6 is made from another sugar (see Tambasco-Studart et al 2005 PNAS). PlantCyc also shows B6 proceeding from R5P, not DXP (https://pmn.plantcyc.org/PLANT/NEW-IMAGE?type=PATHWAY&object=PWY-6466&detail-level=3). I am unaware of other roles for DXP in plants besides the MEP pathway, but authors may cite them if known.
4. I do not see table S1 or S2 with primer sequences.
5. Can SDS-PAGE gels be included as supporting figures?
6. The first kcat in tables 1 and 2 is capitalized but probably shouldn’t be
7. In general, references to supporting figures need to be re-organized. There is a mentione of figure S1 on line 325. The next supporting figure call out is to figure S12 in line 496, followed by S18 in line 506. I do not see the supporting figures as an attachment to the manuscript.
8. Optionally, add individual data points to QPCR data in fig 7. This small addition would improve this figure. More and more journals prefer this format over solid bar graphs, which are slightly misleading. However, the simple statistical analysis presented seems adequate for the larger differences observed, and additional of dots would only enhance the quality of the figure.
Author Response
Responses to Reviewer Comments:
- Authors state “…crosstalk between these pathways has been described in numerous experimental systems (Hemmerlin et al., 2012).” Can additional references (preferably from the last 5 years) be provided to support this statement?
We appreciate the comment. We added a citation for the last comprehensive investigation of the contribution of the MVA and MEP pathways to different plant terpenoids (Lipko and Swiezewska, 2016). Newer reviews do not seem to address pathway crosstalk comprehensively.
2. Line 55, misspelling in Amelunxen, 1965; Amelunxes et al., 1969
Thank you for catching the typo. The citation has been corrected.
- At line 70, authors state “An isomerization and reduction catalyzed by 1-deoxy-D-xylulose-5-phosphate reductoisomerase generates MEP, the eponymous first committed intermediate of the pathway.” Could the authors qualify this statement as perhaps “…first committed intermediate…in bacteria.”? There is good support DXP is in fact the first committed intermediate in plants as vitamin B6 is made from another sugar (see Tambasco-Studart et al 2005 PNAS). PlantCyc also shows B6 proceeding from R5P, not DXP (https://pmn.plantcyc.org/PLANT/NEW-IMAGE?type=PATHWAY&object=PWY-6466&detail-level=3). I am unaware of other roles for DXP in plants besides the MEP pathway, but authors may cite them if known.
We thank the reviewer for the comment and information we had overlooked. The statement has been corrected in the revised manuscript.
3. I do not see table S1 or S2 with primer sequences.
Tables S1 and S2 were included in the materials we submitted to Plants. It is unclear to us why they were not visible. Both supplementary tables are also included in the revised manuscript.
4. Can SDS-PAGE gels be included as supporting figures?
The revised manuscript now has supporting figures showing SDS-PAGE images.
5. The first kcat in tables 1 and 2 is capitalized but probably shouldn’t be
Table 1 and 2 have been corrected.
6. In general, references to supporting figures need to be re-organized. There is a mentione of figure S1 on line 325. The next supporting figure call out is to figure S12 in line 496, followed by S18 in line 506. I do not see the supporting figures as an attachment to the manuscript.
Thank you for the comment. We read over the manuscript carefully and ensured that figures, tables and supplementary materials are mentioned sequentially. We also double-checked that all Supplementary Materials were included in the revised submission.
7. Optionally, add individual data points to QPCR data in fig 7. This small addition would improve this figure. More and more journals prefer this format over solid bar graphs, which are slightly misleading. However, the simple statistical analysis presented seems adequate for the larger differences observed, and additional of dots would only enhance the quality of the figure.
We appreciate the comment. Dots for individual data points have been added to Figure 7.